# Stem-Cell-Derived β-Like Cells with a Functional PTPN2 Knockout Display Increased Immunogenicity

**DOI:** 10.3390/cells11233845

**Published:** 2022-11-30

**Authors:** Taylor M. Triolo, J. Quinn Matuschek, Roberto Castro-Gutierrez, Ali H. Shilleh, Shane P. M. Williams, Maria S. Hansen, Kristen McDaniel, Jessie M. Barra, Aaron Michels, Holger A. Russ

**Affiliations:** Barbara Davis Center for Diabetes, Department of Pediatrics, University of Colorado School of Medicine Anschutz Medical Campus, Aurora, CO 80045, USA

**Keywords:** autoimmunity, type 1 diabetes, genetic risk, PTPN2, CRISPR/Cas9 knockout, stem-cell-derived pancreatic beta cells, direct differentiation, co-culture, autoreactive TCR transductants

## Abstract

Type 1 diabetes is a polygenic disease that results in an autoimmune response directed against insulin-producing beta cells. *PTPN2* is a known high-risk type 1 diabetes associated gene expressed in both immune- and pancreatic beta cells, but how genes affect the development of autoimmune diabetes is largely unknown. We employed CRISPR/Cas9 technology to generate a functional knockout of *PTPN2* in human pluripotent stem cells (hPSC) followed by differentiating stem-cell-derived beta-like cells (sBC) and detailed phenotypical analyses. The differentiation efficiency of *PTPN2* knockout (PTPN2 KO) sBC is comparable to wild-type (WT) control sBC. Global transcriptomics and protein assays revealed the increased expression of HLA Class I molecules in PTPN2 KO sBC at a steady state and upon exposure to proinflammatory culture conditions, indicating a potential for the increased immune recognition of human beta cells upon differential *PTPN2* expression. sBC co-culture with autoreactive preproinsulin-reactive T cell transductants confirmed increased immune stimulations by PTPN2 KO sBC compared to WT sBC. Taken together, our results suggest that the dysregulation of *PTPN2* expression in human beta cell may prime autoimmune T cell reactivity and thereby contribute to the development of type 1 diabetes.

## 1. Introduction

Classically, type 1 diabetes has been viewed as a disease of the immune system present prior to dysglycemia, with autoimmune biomarkers, islet autoantibodies directed against insulin and, other beta cell proteins [1]. Patients do not present clinical signs of beta cell dysfunction until most of their endogenous insulin production has declined. Many studies have been conducted to try to delay the onset of the clinical disease once islet autoantibodies are present [2,3,4]. However, these efforts are complicated by the considerable heterogeneity within type 1 diabetes, as many genes can contribute to the risk of autoimmunity [5,6,7]. Genome-wide association studies have identified over 81 risk alleles for type 1 diabetes, and many associated with immune function, but approximately two-thirds of risk-alleles represent loci associated with genes expressed in pancreatic beta cells [8,9,10]. Accumulating evidence suggests that beta cells may contribute to their own autoimmune demise, but this is a novel hypothesis and has yet to be thoroughly investigated in the human context. This gap in our knowledge is largely due to the scarcity of human islet donor materials, large variations between donor samples, and difficulties in efficiently manipulating primary human beta cells, preventing detailed phenotypic studies. To address these challenges, we and others have demonstrated the successful generation of glucose-responsive human stem-cell-derived beta-like cells (sBC) from human pluripotent stem cells (hPSC) as a model to further study the beta cell’s role in the development of type 1 diabetes [11,12,13,14]. Much work has been performed by us and others to optimize the final stages of differentiation and the characterization of functional sBCs [12,13,15,16]. Clinical trials are underway to investigate the safety and efficacy of sBC as an abundant cell source for cell replacement therapies to treat type 1 diabetes (NCT02239354, NCT03163511, and NCT02939118) [17,18].

Further research into the effects of high-risk genes associated with type 1 diabetes in human pancreatic beta cells will provide important, novel insights into autoimmune initiation and progression. *PTPN2* is a known high-risk gene associated with type 1 diabetes, and polymorphisms are associated with the development of islet autoimmunity [19] and the earlier onset of clinical diseases [20]. *PTPN2* is involved in the JAK/STAT signaling pathway and is expressed in T cells, B cells, and pancreatic beta cells [9,21,22,23]. *PTPN2* mutations in human B cells are associated with earlier onset type 1 diabetes [24]. It is possible that this combinatorial dysfunction in the immune/beta cell’s interface contributes to the initiation of an autoimmune response. In mice, beta cells with deletions in *Ptpn2* exhibit impaired functional responses when exposed to a glucose challenge [24]. Beta cell responses to viral infections are sensitized by the lack of *Ptpn2* and results in increased apoptosis via Bim3 and demonstrates a direct interaction between type 1 diabetes risk genes and proapoptotic pathways in beta cells [25]. *PTPN2* has also been shown to modulate IFN-γ signal transduction in the beta cell and regulate cytokine-induced apoptosis [26]. Functionally, *PTPN2* deficiency has been shown to exacerbate type I and II interferon-signaling networks [27] and knockdown of *PTPN2* in EndoC-βH1 human-beta-like cells demonstrates an amplification of an inflammatory response via STAT1 phosphorylation. We sought to build upon these findings by investigating the effects of a functional *PTPN2* knockout in sBC using detailed phenotypic analyses and novel co-culture systems by employing autoreactive T cell transductants targeting sBC. In this study, we provide evidence that alterations in *PTPN2* expression prime human beta cells for immune recognition, thereby providing a mechanism that may contribute to accelerated disease development.

## 2. Materials and Methods

### 2.1. hPSC Culture and Differentiation of Stem-Cell-Derived Beta-Like Cells (sBC)

Differentiations were conducted using the previously published protocol of our laboratory [15]. Undifferentiated hPSC Mel1^INS-GFP^ reporter cells [28] were maintained on hES-qualified Matrigel (Corning #354277, Sigma-Aldrich, St. Louis, MO, USA) in mTeSR+ media (STEMCELL Technologies #05826, Vancouver, BC, Canada). We carried out differentiations in sBC in a suspension-based, bioreactor magnetic stirring system (Reprocell #ABBWVS03A-6, #ABBWVDW-1013, #ABBWBP03N0S-6, Beltsville, MD, USA). Confluent hPSC cultures were dissociated into single-cell suspension and incubated with TrypLE (Gibco #12-604-021 Waltham, MA, USA) for 8 min at 37 °C. Detached cells were quenched with mTESR+ media, and cells were counted using MoxiGo II (Orflow). This was followed by seeding 0.5 × 10^6^ cells/mL in mTeSR+ media supplemented with 10 μM ROCK inhibitor (Y-27632, R&D Systems #1254-50) in 3D bioreactors and on a magnetic stirring system set at 60 rpm in a cell culture incubator with 5% CO_2_ to induce sphere formations for 48 h, and the media were changed to an equivalent amount of fresh mTeSR+ media. After 72 h, to induce definitive endoderm differentiation, spheres were collected in a 50 mL Falcon tube, allowed to settle by gravity, washed once with RPMI (Gibco #11-875-093) + 0.2% FBS, and re-suspended in d0 media (RPMI containing 0.2% FBS, 1:2000 100x ITS (Gibco #41400-045), 100 ng/mL Activin A (R&D Systems #338-AC-01M), and 3 μm CHIR99021 (STEMCELL Technologies #72054)). Differentiation media were changed daily by letting spheres settle by gravity for 3–10 min. Most supernatants were removed by aspiration; fresh media were added, and bioreactors were placed back on the stirrer system. sBC differentiation was based on our previously published protocol [11] with modifications as outlined below. The differentiation media are as follows: day 1–2 (d1–2): RPMI containing 0.2% FBS, 1:2000 ITS, and 100 ng/mL Activin A; d3–4: RPMI containing 2% FBS, 1:1000 ITS, and 50 ng/mL KGF (Peprotech #100-19-1MG); d5: DMEM with 4.5 g/L D-glucose (Gibco #11960-044) containing 1:50 N21 (R&D Systems AR008), 1:100 NEAA (Gibco #11140-050), 1 mM Sodium Pyruvate (Gibco #11360-070), 1:100 Penicillin Streptomycin (Thermo Fisher Scientific 15-140-122), 3 nM TTNPB, (R&D Systems #0761), 250 nM Sant-1 (R&D Systems #1974), 250 nM LDN (STEMCELL Technologies #72149), 30 nM PMA (Sigma Aldrich #P1585-1MG), 50 μg/mL 2-phospho-L-ascorbic acid trisodium salt (VitC) (Sigma #49752-10G); d6: DMEM with 4.5 g/L D-glucose (Gibco #11960-044) containing 1:50 N21 (R&D Systems AR008), 1:100 NEAA (Gibco #11140-050), 1 mM Sodium Pyruvate (Gibco #11360-070), 1:100 Penicillin Streptomycin (Thermo Fisher Scientific 15-140-122), 3 nM TTNPB, (R&D Systems #0761), 50 μg/mL 2-phospho-L-ascorbic acid trisodium salt (VitC) (Sigma #49752-10G); d7: DMEM with 4.5 g/L D-glucose (Gibco #11960-044) containing 1:50 N21 (R&D Systems AR008), 1:100 NEAA (Gibco #11140-050), 1 mM Sodium Pyruvate (Gibco #11360-070), 1:100 Penicillin Streptomycin (Thermo Fisher Scientific 15-140-122), 100 ng/mL EGF (R&D Systems #236-EG-01M), 50 ng/mL KGF, 50 ug/mL 2-phospho-L-ascorbic acid trisodium salt (VitC) (Sigma #49752-10G); d9–14: DMEM containing 2% fraction V BSA, 1:100 NEAA, 1 mM Sodium Pyruvate, 1:100 ITS, 1:100 P/S, 10 μg/mL Heparin (Sigma #H3149-250KU), 2M N-Acetyl-L-cysteine (Cysteine) (Sigma #A9165-25G), 10 μm Zinc sulfate heptahydrate (Zinc) (Sigma #Z0251-100g), 1x BME, 10 μm Alk5i II RepSox (R&D Systems #3742/50), 1 μm 3,3′,5-Triiodo-L-thyronine sodium salt (T3) (Sigma #T6397), 0.5 μm LDN, 1 μm Gamma Secretase Inhibitor XX (XXi) (AsisChem #ASIS-0149); d15–23: CMRL (Gibco #11530-037) containing 2% BSA, 1:100 NEAA, 1 mM Sodium Pyruvate, 1:100 P/S, 10 μg/mL Heparin, 2 mM Cysteine, 10 μm Zinc, 1x BME, 10 μm Alk5i II RepSox, 2 μm T3, 50 μg/mL VitC, and 1:500 NaOH to adjust pH to ~7.4. Media were changed every other day starting on d9. VitC was freshly administered every day starting on d15.

### 2.2. CRISPR-Cas9 Genome Engineering—PTPN2 KO

hPSC Mel1^INS-GFP^ cells were dissociated into single cells using TrypLE incubation at 37 °C for 8 min. Cells were then quenched with mTeSR+ media and counted using a MoxiGo II cell counter; 2 × 10^6^ cells were transferred into microcentrifuge tubes and washed twice with PBS. Washed cells were then prepared for a CRISPR-Cas9-mediated *PTPN2* gene knockout (Figure 1A, Appendix A). Four guide RNA (gRNA) plasmids, two targeting exon 5 and two targeting exon 7 of the PTPN2 gene, were generated in house and nucleofected along with CRISPR-Cas9 using the following gRNAs: PTPN2 gRNA 5.1 58290-58309 FWD:AGGTTAAATGTGCACAGTAC, PTPN2 gRNA 5.2 58316-58335 REV: AGCATCTCTTGGTCATCTGT, PTPN2 gRNA 7.1 69960-69979 REV: TGAGAATCTCAGTTGATCTG, PTPN2 gRNA 7.2 69962-69981 REV: TATGAGA ATCTCAGTTGATC (Appendix A). Twenty-four hours after plating, cells were selected for 48 h with puromycin (0.5 μg/mL). Individual clonal colonies were picked and amplified for further characterization. Genomic DNA was extracted, and PCR analysis was conducted on exon 5 and exon 7 of *PTPN2* for the WT or KO clones to identify PTPN2 KO clones (Figure 1B,C, Appendix A). The Sanger sequencing for the PCR products was employed to further confirm the PTPN2 KO of clonal hPSC Mel1^INS-GFP^ cell lines (Appendix A).

### 2.3. Immunofluorescence

sBCs were fixed for 9 min at room temperature with 4% paraformaldehyde and then washed twice with PBS. Fixed clusters were then prepped for embedding and cryo-sectioning as follows: Fixed clusters were incubated overnight in 30% sucrose (Sigma #S0389) before embedding in tissue-tek OCT (Sakura #4583) and snap freezing and storing at −80 °C. OCT-blocks containing fixed clusters were cryo-sectioned (10 μm thickness) and transferred to glass slides. Blocking and staining of the cryo-sections proceeded with blocking for 30 min in CAS-block (Thermo Fisher #008102) with 0.4% Triton X-100 (Thermo Fisher #85111) and then incubation in the primary antibody solution (antibody diluted in CAS-block, 0.4% Triton X-100) (Appendix A) occurred overnight at 4 °C. On the following day, the clusters were washed 3 times for 5 min in PBS-containing 0.1% Tween-20 (PBS-T) (Sigma #P4417) and incubated in appropriate secondary antibody solutions (antibody diluted in PBS-T and DAPI (1:1000)) for 1 h at room temperature. The clusters were then washed 2 times for 5 min in PBST and 1 time for 5 min in PBS and mounted with ProLong™ Gold Antifade Mountant with DAPI (Invitrogen, P36935) on glass slides. Antibody dilutions were prepared as indicated in Appendix A. Images were acquired using confocal microscopy (Carl Zeiss LSM 800). The immunofluorescence staining of somatostatin and glucagon cells was quantified by counting positive cells out of all cells identified via Dapi staining. Results were compared using Student’s *t*-test.

### 2.4. Flow Cytometry

hPSC and differentiating clusters were collected and dissociated using 0.05% trypsin/EDTA (Lonza #cc3232) in a 37 °C bead bath (Thermo Scientific, Waltham, MA, USA) for 10 min, pipetted up and down using a p1000 pipette, and quenched with 2% FBS in PBS before centrifugation at 1500 RPM for 3 min. Single cell suspensions were fixed for 8 min at room temperature with 4% paraformaldehyde and then washed twice with PBS. Clusters were stained for pluripotency markers, definitive endoderm markers, viability, and HLA-ABC surface markers using the antibodies indicated in Appendix A. Clusters collected for HLA-ABC surface-marker staining were not fixed with 4% paraformaldehyde but stained live. sBCs were analyzed at a steady state and after exposure to IFNγ at 100 ng/mL for 48 h. Single cell suspensions were filtered through 40 μm cell strainers into 5 mL FACS tubes and stained for 20 min on ice for marker proteins. After incubation, the cells were washed with a FACS buffer (PBS containing 2% FBS and 2 mM EDTA) and strained again through a cell strainer and resuspended in a FACS buffer for analyses on CYTEK Aurora. Antibody dilutions were prepared as indicated in Appendix A. Analysis and graphs were made using FlowJo software v10.6.2. For gating, live cells were selected based on forward scatter vs. side scatter, and single cells were selected based on the side scatter area vs. side scatter height. A viability dye was used to delineate between viable and dead cells. Then, HLA-ABC-positive cells were selected from both all and GFP+-gated cells.

### 2.5. GFP+ sBC Sorting

sBC clusters were collected in an Eppendorf tube, allowed to settle by gravity, and then the supernatant was removed; the clusters were washed once with PBS. Clusters were dissociated in 0.05% trypsin/EDTA (Lonza #cc3232) in a 37 °C bead bath (Thermo Scientific, Waltham, MA, USA) for 15 min. The clusters were dissociated and then quenched with 2% FBS in PBS. The suspension was spun down at 1500 RPM for 3 min. The supernatant was removed, and the cells were resuspended in a FACS buffer. Cells were filtered through a 40 μm cell strainer into 5 mL tubes (Falcon #352235). GFP+ sBCs were analyzed on a BioRad S3e Cell Sorter and gated for GFP on the 488/FITC channel.

### 2.6. Content Analysis

Total insulin and proinsulin content analyses were carried out on aliquots of 1000 fluorescence-activated cell sorting (FACS)-sorted GFP+ sBC single cells lysed in acid ethanol using commercially available ELISA kits (insulin: Alpco 80-INSHU-E01.1; proinsulin: 80-PINHUT-CH01).

### 2.7. Western Blotting

Total protein was extracted from ~1 × 10^5^ cells using RIPA lysis buffer. The lysis buffer was supplemented with EDTA-free protease inhibitor (Roche 4693159001). Lysates were resolved on SDS–PAGE, transferred to a methanol-activated PVDF membrane (BioRad, 1620177), blocked in 5% milk for 30 min, and followed by incubation with the indicated primary antibodies (Appendix A) overnight at 4 °C. After washing, membranes were then incubated for 1hr at room temperature with the secondary antibody conjugated to horseradish peroxidase (HRP), as indicated in Appendix A. After incubation with Clarity Western ECL Substrate (BioRad, 1705061), bands were detected with an Azure Biosystems Western blotting imager.

### 2.8. Bulk RNA Sequencing

Total RNA was isolated from 50,000 sorted GFP+ sBCs cells using RNEasy kits from Qiagen. Sequencing libraries were generated using the NEBNext Ultra II Directional RNA Library kit with NEBNext rRNA depletion. Paired-end sequencing reads were trimmed using cutadapt (v1.16) [29] and aligned using STAR (v 2.5.2a) [30], and exonic read counts were quantified using featureCounts from the subread package (v1.6.2) [31]. Differentially expressed genes were identified using DESeq2 (v1.24.0) [32]. Heatmaps were generated using ComplexHeatmap and ordered using the hierarchical clustering of Euclidean distances with the complete method [33] RNA-seq data-processing and analyses were performed with Pluto data analysis software (https://pluto.bio accessed on 27 April 2022).

### 2.9. Differential Expression Analysis

Differential expression analysis was performed when comparing the groups: WT vs. PTPN2 KO. Genes were filtered to include only genes with at least 3 reads counted in at least 20% of samples in any group. Differential expression analyses were then performed with the DESeq2 R package, which tests for differential expressions based on a model using the negative binomial distribution. Log_2_ fold changes were calculated for the comparison: WT vs. PTPN2 KO. Thus, genes with a positive log_2_ fold change value had increased expressions in PTPN2 KO samples. Genes with a negative log_2_ fold change value had increased expressions in WT samples. The false discovery rate (FDR) method was applied for multiple testing correction. FDR-adjusted *p*-values are shown on the *y*-axis of the volcano plot. An adjusted *p*-value of 0.001 was used as the threshold for statistical significance.

### 2.10. Gene Set Enrichment Analysis (GSEA)

Gene set enrichment analysis (GSEA) was performed using the fgsea R package and the fgseaMultilevel() function [31]. The fold-change from the WT vs. PTPN2 KO differential expression comparison was used to rank genes. Hallmark gene set collections from the Molecular Signatures Database (MsigDB) [32,33] curated using the msigdbr R package SEA results for each tested gene set are shown in the table. The Adj *p*-value column contains the false discovery rate (FDR)-adjusted *p*-value. The NES column contains the normalized enrichment score (NES) computed by GSEA, which represents the magnitude of enrichment as well as the direction. A positive NES indicates more enrichment in the first group, while a negative NES indicates more enrichment in the second group.

### 2.11. T Cell Stimulation Assay

T cell stimulation assays were conducted as previously reported by our laboratory [34]. sBC clusters were washed with PBS and incubated with 0.05% Trypsin with EDTA at 37 °C for 12 min to create a single cell suspension and quenched with 2% FBS in PBS. Cells were counted and plated into Matrigel-coated 96 well plates and incubated for 48 h for T cell stimulation assays or in 6 well plates for parallel flow cytometry characterization [34]. After 48 h, the cultures were washed twice with PBS and incubated with PPI: 15–24 (ALWGPDPAAA) at 10 µg/mL or PPI: 1–11 (MALWMRLLPLL) peptide at 10 µg/mL for 4 h [35]. The insulin peptides used for these stimulation assays were obtained from Genemed Synthesis Inc. at >95% purity and dissolved in PBS at a neutral pH. 1 × 10^5^ 5KC-1.E6 T cells (TCR transductant responding to PPI: 15–24 presented by HLA-A*02:01) or 5KC-1.C8 T cells (TCR transductant responding to PPI: 1–11 presented by HLA-A*02:01) were co-cultured with sBC overnight. T cells of the 5KC type were cultured without sBC or the peptide acted as a negative control, and those treated with anti-CD3 monoclonal antibody (eBioscience, clone 2C11) at a concentration of 10 µg/mL acted as a positive control for each experiment. Murine IL-2 secreted by 5KC T cells was measured in the culture supernatant using a highly sensitive ELISA (V-PLEX IL-2 kit, Meso Scale Diagnostics, LLC, Rockville, MD, USA) followed by detection on the MESO QuickPlex SQ120 instrument.

### 2.12. Statistical Analysis

All statistical analyses were performed in GraphPad Prism (Version 9.0.0) or Pluto data analysis software. Student’s *t* test was used for samples with 2 groups. For multiple group analyses, one- or two-way ANOVAs were used with the suggested multiple comparison tests.

### 2.13. Data and Resource Availability

Generated and analyzed data sets from the current study were uploaded to GEO GSE211636 (https://www.ncbi.nlm.nih.gov/geo/query/acc.cgi?acc=GSE211636). URL accessed on 31 August 2022. Reviewer token: edclimewvdotvkh.

## 3. Results

### 3.1. Generation of a CRISPR-Cas9-Mediated PTPN2 Knockout hPSC

To study the role of *PTPN2* in sBC, we used a previously reported hPSC Mel1^INS-GFP^ line in which a GFP reporter is driven by the endogenous insulin promoter [28]. A CRISPR-Cas9-mediated gene knockout was performed at the functional domain of *PTPN2* [36] between exon 5 and 7 with two gRNAs at each exon in plasmids that constitutively express Cas9 and a puromycin gene (Figure 1A, Appendix A). Clonal lines were established by puromycin selection followed by expansion and downstream testing for successful *PTPN2* knockout generation. Clonally derived unmodified stem cell lines served as controls for this study. Genomic DNA PCR analyses identified three clones of interest (Appendix A). The Sanger sequencing of PCR products demonstrated a large missense mutation in exon 5 in clone 8 (Appendix A), a frameshift deletion in clone 19, and an in-frame deletion in clone 28. We selected clone 8, referred to as PTPN2 KO, for further experimentation. Using primers spanning the wild-type (WT) sequence at exon 5 showed a PCR amplification band in WT control cells but not in the PTPN2 KO cells (Figure 1B, Appendix A). Conversely, by using primers spanning exon 5 and exon 7, resulting in PCR amplifications only after the successful knockout of the functional domain of *PTPN2*, we observed a prominent band in the PTPN2 KO clone but no band in the WT control cells (Figure 1C, Appendix A). The Sanger sequencing of the amplified band further verified a large deletion between exon 5 and 7 at the expected bases next to the PAM region of the employed gRNAs (Appendix A). Immunofluorescence (IF) analysis verified that PTPN2 KO cells retained the expression of key pluripotency markers OCT4, NANOG, and SOX2 at comparable levels to WT cells (Figure 1D). Frozen master stocks were established to allow conducting experiments using cells with comparable culture passage numbers. The karyotypic analysis of PTPN2 KO master stock cells revealed no detectable clonal abnormalities of the chromosome number or structure in 18 of 20 cells assayed. However, there were two tetraploid cells, and the loss of chromosome 18 in one cell and the loss of chromosome 22 in a second cell were also detected. These results do not meet the definition of a clone and are considered single-cell anomalies. The Western blot analysis of WT and PTPN2 KO clonal lines (Figure 1E, Appendix A) demonstrate PTPN2 protein expression in the WT cell line but not in the PTPN2 KO line verifying the genomic DNA PCR results. Taken together, we confirmed the successful disruption of *PTPN2* while successfully maintaining pluripotency in hPSCs.

### 3.2. PTPN2 KO hPSC Efficiently Differentiate into Pancreatic Stem-Cell-Derived Beta Cells

WT and PTPN2 KO cell lines underwent a suspension culture-based direct differentiation protocol to generate sBCs based on our previous work (Figure 2A) [11,15]. WT and PTPN2 KO maintained similar morphological appearances throughout key developmental stages, including the live imaging of the GFP reporter expression at the sBC stage (Figure 2B). The flow cytometry quantification of pluripotency markers SOX2 and TRA160 and definitive endoderm markers FOXA2 and SOX17 similarly did not show differences between WT and PTPN2 KO cells (Figure 2C, Appendix A). The IF analysis of WT and PTPN2 KO sBC clusters for beta cell markers PDX1, insulin, and C-peptide appeared comparable between samples (Figure 2D). Quantification somatostatin and glucagon did not reveal statistically significant differences between WT and PTPN2 KO sBCs. The live imaging and flow-based quantification of insulin expression using the GFP reporter revealed, on average, 14% GFP^+^ sBC with no significant differences detected between the experimental groups (Figure 2E,F). The FACS-sorted aliquots of 1000 GFP^+^ sBC from either WT and PTPN2 KO were quantified for total proinsulin and insulin content using commercially available ELISA kits and did not show significant differences between sBC generated from either line (Figure 2G). Of note, the levels of proinsulin and insulin content were similar to our previous studies despite the percentage of sBC generated being lower than that reported previously [15]. Overall, throughout key developmental stages of the differentiation protocol, the generation of WT and PTPN2 KO sBC was similar. However, the Western blot analysis of GFP^+^ sorted sBC demonstrates the PTPN2 protein in WT but not in PTPN2 KO sBCs (Figure 2H, Appendix A). Together, these data provide evidence for the generation of an appropriate model to study the impact of *PTPN2* disruption in human sBCs.

### 3.3. PTPN2 KO sBC Display Increased HLA Class I Molecules Compared to WT sBCs

We investigated global gene expression changes using the bulk-RNA sequencing of single-cell FACS-sorted GFP^+^ sBC from four independent differentiation experiments using WT and PTPN2 KO cells. The differential gene expression analysis indicates 308 differentially expressed genes, with 187 upregulated and 121 downregulated (*p* < 0.01) between the WT and PTPN2 KO samples (Figure 3A, Appendix A). Of note, key beta cell genes were not differentially expressed between samples (Appendix A). Expressions between exon 5 and 8 of *PTPN2* RNA are absent in PTPN2 KO samples compared to WT samples, while transcriptions from exons 1 to 4 remain intact, further verifying targeted gene editing at the expected genomic location in this clone (Appendix A). Of interest, IRAK4 was significantly downregulated in PTPN2 KO sBCs compared to WT sBCs. IRAK4 is a protein kinase involved in signaling innate immune responses from TLR and codes for a kinase that activates NFKB in both TLR and TCR signaling, which is important for innate immune responses against foreign pathogens [37,38,39]. Conversely, IL17RA and IL1R1 were both upregulated in the PTPN2 KO sBCs, both of which are involved in NFKB signaling. Additionally, factors associated with MHC Class I-mediated antigen processing including BCAP31, NFE2L2, MYLIP, VAMP8, ITCH, UBE2W, and SEC24D were upregulated in PTPN2 KO sBCs. HM13 is required to generate lymphocyte cell surface (HLA-E) epitopes derived from MHC-I signal peptides and was also upregulated in PTPN2 KO sBCs.

The gene set enrichment analysis shows the significant upregulation of hallmark interferon gamma response genes, oxidative phosphorylation, and IL-6, JAK, and STAT3 signaling (Figure 3B–D) in the PTPN2 KO cells compared to WT, thereby verifying recent results showing the upregulation of interferon gamma signaling after *PTPN2* knock down in different beta cell models [27].

While not reaching significance, we noticed an upregulation in HLA class I transcript in PTPN2 KO cells compared to WT controls and conducted additional analyses (Appendix A). WT and PTPN2 KO sBC clusters were stained and quantified using flow cytometry for surface HLA Class I molecule expression (Figure 3E). At a steady state, HLA Class I positive cells significantly increased from an average of 21% on WT controls to 55% on PTPN2 KO cells. Gating specifically on GFP^+^ sBC revealed a significant increase in HLA class I^+^ cells from on average 14% in WT controls to 62% in PTPN2 KO cells (*n* = 7, Figure 3E,G,H). sBC exposed to the proinflammatory cytokine IFN-γ at 100 ng/mL (*n* = 3) showed HLA class I expression on 35.73% of WT cells and 69.83% on PTPN2 KO cells. Gating specifically on GFP^+^ sBC similarly revealed a significant increase in HLA Class I-expressing cells from 53.80% on WT controls to 83.33% on PTPN2 KO cells (*n* = 7 untreated and *n* = 3 treated, Figure 3F–H). Increased HLA Class I surface expression was also apparent on PTPN2 KO compared to WT hPSC with and without treatments with IFN-γ (Appendix A). IF and Western blot analyses confirmed the findings of increased HLA class I protein expression on PTPN2 KO sBC compared to WT controls (Figure 3I,J).

### 3.4. PTPN2 KO sBC Display Increased Stimulation of Autoreactive TCR Transductants Compared to WT sBC

To test if PTPN2 KO sBC could be more readily recognized by preproinsulin-specific CD8 T cell receptors (TCRs), we employed our previously reported T-cell transductant sBC co-culture system [34]. This system provides a sensitive approach to test for the immune stimulation of autoreactive human TCRs isolated from T cells found within the residual islets from type 1 diabetes organ donors [40] (Figure 4A). Briefly, identified human TCR α/β genes are retrovirally expressed in an immortalized mouse T cell devoid of endogenous TCR expression, thus creating a human-specific TCR transductant that secretes mouse interleukin-2 (IL-2) upon stimulation with cognate peptide/MHC [34]. We tested the stimulation of two autoreactive CD8 TCRs specific for preproinsulin PPI 1–11 (TCR #1) or PPI 15–24 (TCR #10) [40] peptides, both restricted to HLA-A*02:01. We co-cultured either transductants with the WT control or PTPN2 KO d23 sBC, both expressing human HLA-A*02:01 and facilitating matched cell–cell interactions. T cell transductants alone or treated with anti-CD3/CD28 antibodies served as negative and positive controls, respectively. Using five independent differentiation experiments followed by co-culture revealed a significantly higher stimulation of autoreactive TCR transductants by PTPN2 KO sBC cultures compared to WT controls (TCR #1, *n* = 1, and TCR #10 *n* = 3) (Figure 4B,C). These results provide support for the intriguing concept that *PTPN2* expression regulates the immunogenicity of beta cells via the differential expression of HLA class I molecules.

## 4. Discussion

In this study, we have shown that dysfunctions in *PTPN2* expression in human beta cells has subtle yet potentially important effects on key proteins involved in the immune recognition of beta cells. These findings are among the first to provide insights into how type 1 diabetes risk genes might increase the propensity of genetically at-risk individuals to develop autoimmune diabetes. Understanding the role of genetic dysfunction in distinct cell types contributing to type 1 diabetes risk will further enhance our understanding of disease pathogenesis and aid in identifying novel therapeutic targets aimed at treating the underlying autoimmunity in type 1 diabetes.

A particular understudied aspect of type 1 diabetes pathogenesis is the immune-beta cell interface in a human context due to distinct experimental challenges. Human cadaveric islets containing insulin-producing beta cells exhibit considerable variability due to donor and procurement differences and have been difficult to manipulate effectively. Here, we took advantage of our expertise in generating sBCs as a relevant model system that can provide functional human beta cells in a controlled and reproducible manner. While two non-coding risk alleles for *PTPN2* have been associated with type 1 diabetes, we generated a functional knockout of the gene using CRISPR-Cas9 technology to probe for the effects of altered *PTPN2* gene expression in human beta cells. Importantly, we could reproduce aspects of previous studies (that used a similar knockout approach in mouse and human beta cell models) and found an upregulation of IFN-γ response genes in mutant beta cells [27]. This upregulation may sensitize beta cells and their microenvironment to stress, thereby potentially enhancing the immune-mediated targeting of pancreatic beta cells.

Indeed, previous studies showed that beta cells without *PTPN2* expression have impaired function and displayed increased levels of apoptosis upon stress exposure, including mimicking viral infections and transplantation [25,26]. We did not detect expression changes in key beta cell genes that could explain the observed defects in insulin secretory capacity, as described by others [24]. However, we did notice a change in HLA class I transcript expression levels, but this did not reach statistical significance. A further detailed analysis of protein levels using complementary assays convincingly demonstrates an upregulation of HLA class I molecules on the surface of PTPN2 KO sBC under steady state and upon proinflammatory cytokine exposure mimicking type 1 diabetes. Other non-sBC cells present in differentiated clusters followed a similar trend, suggesting a broad effect of *PTPN2* modulation on HLA class I expression. A recent study showed that cancer cells can be sensitized for effective immunotherapy by sgRNA CRISPR-Cas9-mediated knockout of *PTPN2*, resulting in the upregulation of HLA class I surface molecules, similarly to what we observed for beta cells [41]. RNAseq data suggest that the expressions of factors associated with MHC Class I regulation were upregulated in PTPN2KO sBCs, and the expressions in the NFκB signaling pathway were both up- and downregulated. It is possible that IL1 receptor levels are still intact via MAPK signaling pathways and much work has been conducted by others to understand the relationship between *PTPN2* and inflammatory pathways [27,41,42,43,44]. Given that this gene expression data is not at the functional protein level, future analyses could focus on investigating the relationship between *PTPN2* and these pathways in this model.

To assess the effects of *PTPN2* knockout on immune recognition more directly, we took advantage of our recently published co-culture system by employing T cell receptor transductants [34]. Transductants express human T cell receptors identified from T cells present in the islets of type 1 diabetes organ donors. The two TCR transductants employed in this study are restricted to preproinsulin peptides presented by HLA-A2 molecules. HLA-A2 is also expressed by sBC, providing an HLA–peptide–TCR-matched immune beta cell interaction model. Using this sophisticated approach, we could show that autoreactive TCRs are more readily stimulated by PTPN2 KO sBC compared to the WT controls. These experiments provide evidence for a novel mechanism by which altered *PTPN2* expression in beta cells might contribute to the increased risk for autoimmune beta cell targeting. To further refine our findings, future work will investigate the correlation of known risk SNPs in the *PTPN2* loci and expression levels of *PTPN2* and HLA class I molecules. In addition to distinct phenotypes detected in beta cells, dysfunctions in protein tyrosine phosphatases are known to impact antigen receptor signaling and cytokine signaling. This likely contributes to the development of autoimmunity via multiple mechanisms, such as failures of central and peripheral tolerance and the promotion of proinflammatory T cell responses [45]. In addition, the type 1 diabetes-associated risk variant of *PTPN2* rs1893217 independently contributed to diminished IL-2 receptor signaling in patient T-regulatory cells [46]. Taken together, these studies and our presented results indicate that rather than triggering a specific disease-causing phenotype within a single cell type, *PTPN2* polymorphisms might contribute to subtle yet critical dysfunction in multiple cell types, thereby priming an individual for disease development. In sum, our work contributes critical insights into type 1 diabetes pathogenesis that further supports the concept that beta cells contribute to the erroneous immune-mediated destruction by autoreactive T cells.

## Figures and Tables

**Figure 1 cells-11-03845-f001:**
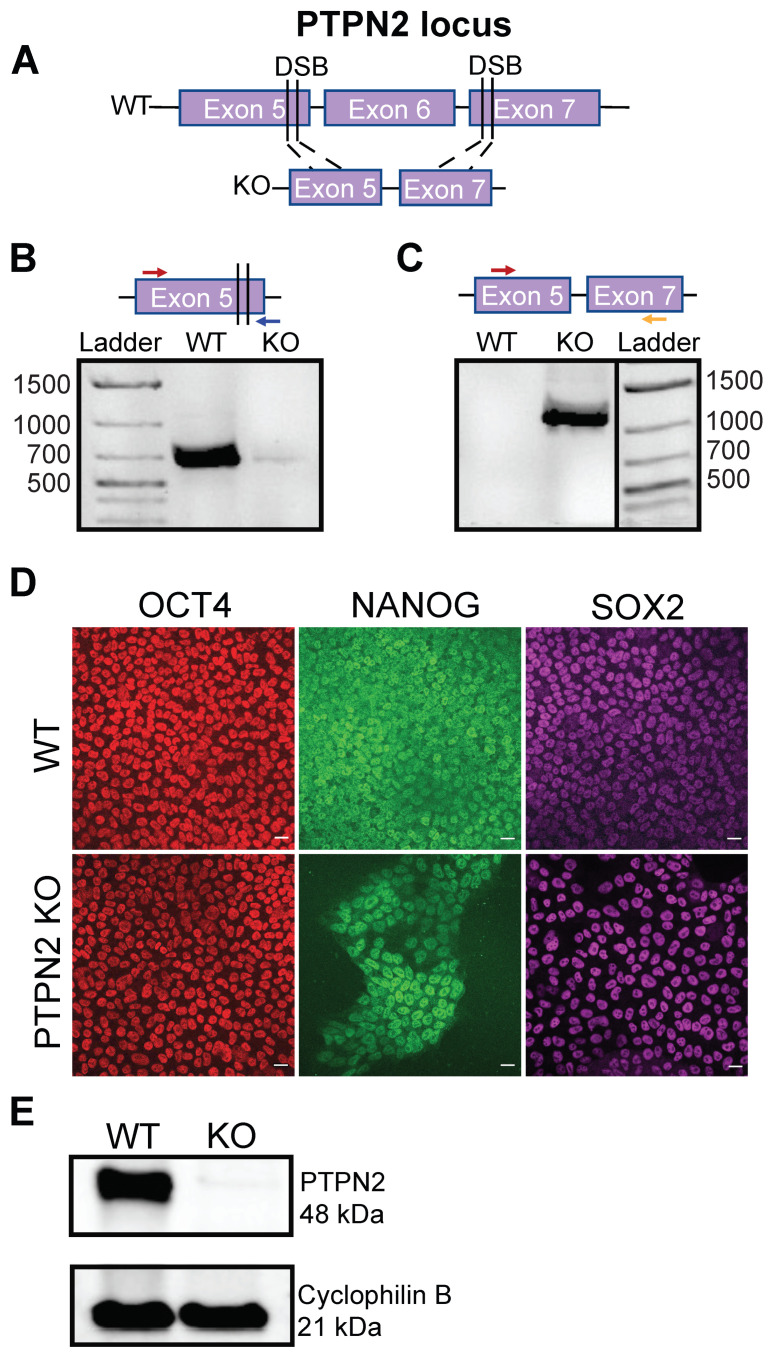
Generation of a CRISPR-mediated PTPN2 knockout in human pluripotent stem cells. (**A**) Schematic of CRISPR-mediated disruption of PTPN2 in hPSCs at the functional domain between exons 5 and 7. Four gRNAs targeted the functional domain of PTPN2, with two at exon 5 and two at exon 7, resulting in a large excision upon successful targeting. (**B**) Representative gDNA PCR amplification of clonal WT and PTPN2 KO hPSC lines employing primers that will only amplify intact, unedited DNA of exon 5. A 639 bp band amplified in WT and no band amplified in the PTPN2 KO (*n* = 2, independent analyses). (**C**) Representative gDNA PCR amplification of clonal WT and PTPN2 KO hPSC lines employing primers that will only amplify after the successful deletion of the DNA segment spanning exon 5–7 with the forward primer in exon 5 and the reverse primer in exon 7. A 1050 bp band amplified in PTPN2 KO but not in WT (*n* = 2, independent analysis). (**D**) Representative immunofluorescence images of WT and PTPN2 KO hPSC colonies. Samples were stained for OCT4 (red), NANOG (green), and SOX2 (purple) (*n* = 2, independent analysis). Scale bar represents 20 μm. (**E**) Representative Western blot analysis of PTPN2 protein expression in WT and PTPN2 KO hPSC lines indicating expression in WT hPSCs at 48 kDa and no expression in PTPN2 KO hPSCs (*n* = 3, independent analysis).

**Figure 2 cells-11-03845-f002:**
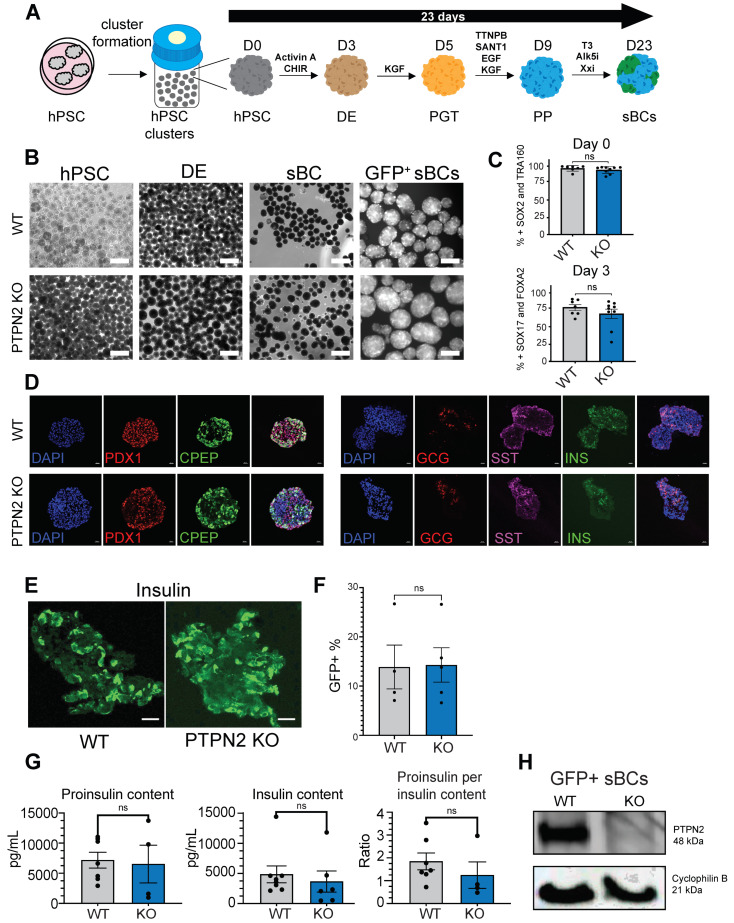
Direct differentiation of WT and PTPN2 KO to generate stem-cell-derived beta-like cells. (**A**) Schematic of stepwise suspension culture based direct differentiation of WT and PTPN2 KO sBC clusters. Critical developmental factors at key stages are shown. Human pluripotent stem cells (hPSC), definitive endoderm (DE), primitive gut tube (PGT), pancreatic progenitor (PP), and stem-cell-derived beta-like cells (sBC). (**B**) Representative bright field images of WT and PTPN2 KO differentiating clusters at key developmental stages. hPSC clusters, DE, and sBC clusters with live fluorescence imaging of insulin promoter-driven GFP expression. Scale bars represent 200 μm for the hPSC and DE images and 600 μm for sBC and GFP+ sBC images. (**C**) Flow cytometric quantification of pluripotency markers at day 0 (SOX2 and TRA160) and DE markers at day 3 (SOX17 and FOXA2) for WT (gray) and PTPN2 KO (blue) clusters. (*n* = 7 WT, *n* = 9 KO) ns = no significant difference, (**D**) Representative immunofluorescence images of WT and PTPN2 KO sBC clusters. Samples were stained for DAPI (blue), PDX1 (red), C-peptide (green), glucagon (red), somatostatin (violet), and insulin (green). Scale bar represents 20 μm. (*n* = 3 WT, *n* = 2 PTPN2 KO). (**E**) Representative immunofluorescence images of WT and PTPN2 KO sBC clusters. Samples were stained for insulin (green). Scale bar represents 20 μm (*n* = 3 for WT and PTPN2 KO). (**F**) Quantification of GFP expression in WT and PTPN2 KO sBC. Data are presented as mean +/− SE percentage of GFP+ cells from 4 independent differentiations. Analyzed with unpaired Student’s *t* test with no significant difference denoted as ns. (**G**) Total proinsulin content, insulin content, and proinsulin-to-insulin content ratios per 1000 FACS-sorted GFP+ sBC (*n* = 7 WT and *n* = 4 PTPN2 KO independent differentiation experiments with 3 × 1000 cells collected per experiment). Error bars are representative of the mean ± SE. Analyzed with unpaired Student’s *t* test with no significant differences denoted as ns. (**H**) Representative Western blot analysis for the PTPN2 protein or endogenous control protein Cyclophilin B of sorted day 23 GFP+ sBCs from WT and PTPN2 KO (*n* = 3 independent differentiations).

**Figure 3 cells-11-03845-f003:**
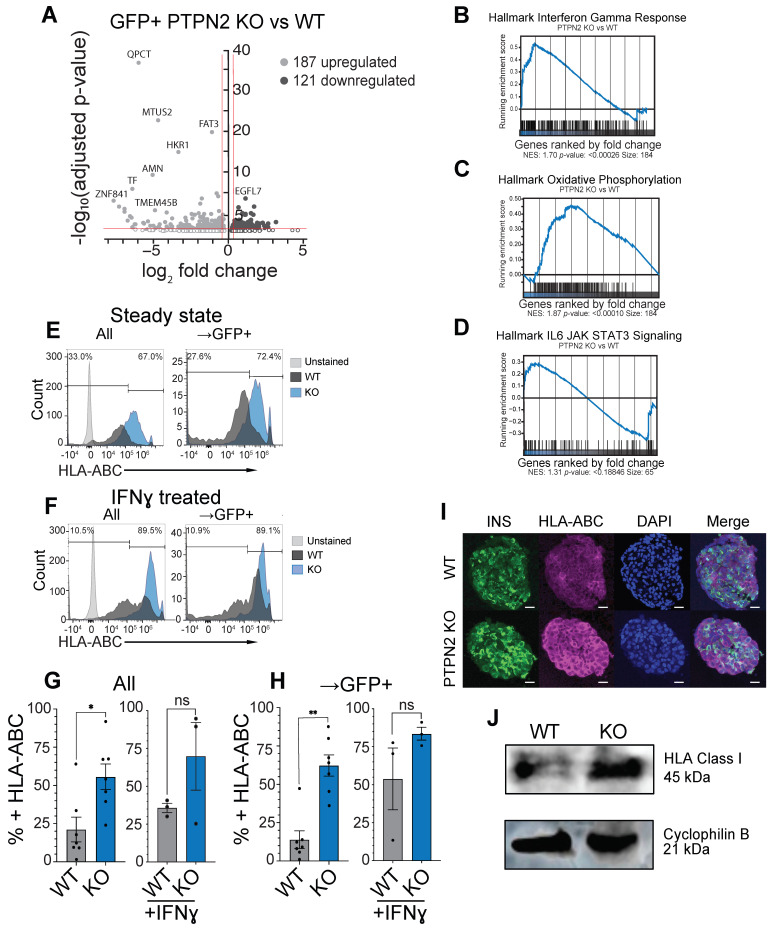
PTPN2 KO sBC displayed increased HLA class I expression. (**A**) Volcano plot of bulk RNAseq differential gene expression of GFP+ WT and PTPN2 KO sBCs. An adjusted *p*-value of 0.01 was used as the threshold for statistical significance with 308 differentially expressed genes. (**B**–**D**) Gene set enrichment from the bulk RNA seq of GFP+ WT and PTPN2 KO sBCs of differentially expressed genes reveals hallmark IFNγ response (**B**), oxidative phosphorylation (**C**), and IL6 JAK/STAT3 signaling (**D**). Ranked genes are shown along the *x*-axis with the vertical ticks representing the location of the genes in this gene set. The heatmap displays the expression of the genes: right showing those more expressed in the first group (PTPN2 KO) and left showing those more expressed in the second group (WT). The blue line shows the enrichment score. (**E**,**F**) Representative flow cytometry histogram and quantification (**G**,**H**) for the detection of surface HLA-ABC expression in WT (black) and PTPN2 KO sBC (blue) at steady state (*n* = 7) or treated with IFNγ for 48 h with 100 ng/mL (*n* = 3) shown with sBC clusters and gated for GFP+ sBCs. (* *p* ≤ 0.03, ** *p* ≤ 0.002). (**I**) Representative immunofluorescence image (*n* = 2) of WT (top) and PTPN2 KO (bottom) sBCs expressing insulin (green) HLA-ABC (pink) and DAPI (blue). Scale bar represents 20 μm. ns = no significant difference (**J**) Western blot of proteins extracted from GFP+ WT and PTPN2 KO sBCs expressing HLA Class 1 (*n* = 1).

**Figure 4 cells-11-03845-f004:**
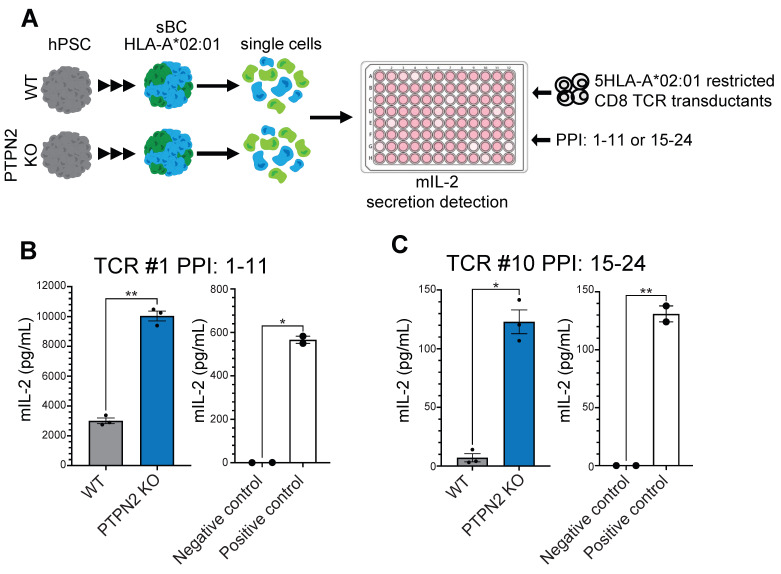
Activated autoreactive T-cell transductants produce increased levels of IL-2 when stimulated by PTPN2 KO sBCs. (**A**) Schematic depicting the sBC and CD8 T cell receptor transductant (5KC-1.E6 or 5KC-1.C8) co-culture experiment. (**B**,**C**) Representative stimulation assay (of 3 technical repeats) of WT (gray) and PTPN2 KO (blue) sBC co-cultured with 5KC-1.C8 (TCR #1 PPI: 1-11) (*n* = 1) (**B**) or with 5KC-1.E6 (TCR #10 PPI: 15-24) (*n* = 4) transductants cells. T cells of type 5KC alone or treated with anti-CD3 antibody served as negative and positive controls, respectively. Data are presented as mean secreted mIL-2 concentration +/− SEM. Unpaired *t*-test was performed with * *p* ≤ 0.0332 and ** *p* ≤ 0.0021.

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
