# Peer review of "Stem-Cell-Derived β-Like Cells with a Functional PTPN2 Knockout Display Increased Immunogenicity"

_cells, 2022, doi:10.3390/cells11233845_

Round 1

Reviewer 1 Report

In the present study, the authors took advantage of human pluripotent stem cells to investigate the effects of knocking out PTPN2, a type 1 diabetes candidate gene. The PTPN2 KO and control stem cells were differentiated in beta-like cells (sBC) and was observed no differences in insulin-positive cells. The authors also performed global transcriptomics and protein assays and found increased expression of HLA Class I molecules in PTPN2 KO beta like-cells. Overall, the results are confirmatory and in line with previous published studies. Some important points should be addressed by the authors.

Major points:

1)    The authors should include the quantification of glucagon and somatostatin positive cells after differentiation.

2)    The quality of the Western blot images should be improved, the blots are overexposed.

3)    Are the genes identified in the RNA-Seq dataset by the authors (Fig. 3A) associated to PTPN2 in other studies? It is important to provide protein validation of key targets (e.g., IRAK4).

4)    Is NF-KappaB signaling affected by PTPN2 deletion?

5)    PTPN2 regulates TNF-alpha signaling in Crohn’s disease (Schart M et al 2011). It will be relevant to study the apoptotic/signaling pathways modulated by PTPN2 in TNF-alpha-treated beta cells in the context of T1D.

Reviewer 2 Report

The authors describe increased HLA Class I expression on PTPN2 KO beta cells that facilitates increased activation of an islet-specific transduced CD8 T cell line.  This finding importantly expands on genotype/phenotype studies and describes a possible functional consequence of variation in PTPN2 expression.  Experiments were well designed and controlled, and interpretation is clear and concise without over-interpreting results from a KO.  Edits are suggested that may improve the manuscript.

1)    Can a CRISPR control be added (either another unrelated gene or empty vector) to control for impact of CRISPR? 

2)    Can increased IFN signaling beyond GSEA be shown to confirm altered function of PTPN2 KO sBC? Possibly other IFN responsive genes following IFN exposure? This becomes important since the emphasis in this paper is on HLA Class I which trended in bulk RNA-seq but did not reach significance

3)    Are there other “trending” genes that also implicate HLA Class I expression or IFN-induced genes? This evidence may help strengthen the rationale for investigating Class I expression with this experimental system. In a similar light, are there other proteins up-regulated with Class I or IFN-regulated that would be a good control for specificity? 

4)    Are there differences in Class I expression on PTPN2 SNP typed hPSC?  Not essential experiment for this manuscript, but worth discussing evidence and implications in the discussion.

5)    Stimulation of CD8 T cells by PTPN2 KO is quite impressive with both transduced cell lines. It is surprising that the anti-CD3 positive control is as high as the experimental control that relies on insulin presentation. A specificity control may help interpret this data (i.e. a line specific for a non-islet antigen). Also, data in B with an n=1 and super high levels of IL-2 with the KO that far exceed the positive control make me wonder about reproducibility and specificity of the stimulation.

Round 2

Reviewer 1 Report

The authors addressed all my comments.

Reviewer 2 Report

authors appropriately addressed critiques